# Spin-filtered measurements of Andreev bound states in semiconductor-superconductor nanowire devices

David van Driel[1,3], Guanzhong Wang[1,3], Alberto Bordin[1], Nick van Loo [1],
Francesco Zatelli[1], Grzegorz P. Mazur[1], Di Xu [1], Sasa Gazibegovic[2],
Ghada Badawy [2], Erik P. A. M. Bakkers [2], Leo P. Kouwenhoven[1] &
Tom Dvir [1,3] ✉

Semiconductor nanowires coupled to superconductors can host Andreev bound states with distinct spin and parity, including a spin-zero state with an even number of electrons and a spin-1/2 state with odd-parity. Considering the difference in spin of the even and odd states, spin-filtered measurements can reveal the underlying ground state. To directly measure the spin of single-electron excitations, we probe an Andreev bound state using a spin-polarized quantum dot that acts as a bipolar spin filter, in combination with a non-polarized tunnel junction in a three-terminal circuit. We observe a spin-polarized excitation spectrum of the Andreev bound state, which can be fully spin-polarized, despite strong spin-orbit interaction in the InSb nanowires. Decoupling the hybrid from the normal lead causes a current blockade, by trapping the Andreev bound state in an excited state. Spin-polarized spectroscopy of hybrid nanowire devices, as demonstrated here, is proposed as an experimental tool to support the observation of topological superconductivity.

Low-dimensional III-V semiconductors proximitized via coupling to superconductors have been researched extensively in recent decades[1,2]. Interest in these hybrid systems is a result of their gate-tunability, strong response to magnetic fields, and large spin-orbit interaction, all combined with superconductivity[3,4]. This makes superconductor–semiconductor hybrids a candidate for creating a topological superconducting phase hosting Majorana zero modes. However, the intrinsic disorder in these systems can lead to localized Andreev bound states (ABSs) that reproduce many of the proposed Majorana signatures[5]. Proximitized InSb nanowires, such as those used in this work, can be tuned between three regimes of superconductor–semiconductor coupling using electrostatic gates[6,7]. When the electron wavefunction is pushed toward the superconductor by negative gate voltage, the nanowire is fully proximitized and the

density of states exhibits a hard superconducting gap. When the electron wavefunction is drawn into the semiconductor away from the superconductor by positive gate voltage, the proximity effect weakens and the density of states becomes gapless, i.e., the gap is soft. In the intermediate regime between these two, a confined hybrid nanowire hosts discrete subgap ABSs, whose electrochemical potential is controlled by gate. It was recently shown that an ABS spanning the entire superconductor-nanowire hybrid length gives rise to non-local transport phenomena[8–12], including equal-spin crossed-Andreev reflection[13], which enables the formation of a minimal Kitaev chain hosting Majorana bound states[14].

A confined semiconductor can host discrete quantum levels. Coupling the semiconductor to a superconductor allows the two to exchange a pair of electrons in a process known as Andreev reflection.

[1]QuTech and Kavli Institute of NanoScience, Delft University of Technology, 2600 GA Delft, The Netherlands. [2]Department of Applied Physics, Eindhoven University of Technology, 5600 MB Eindhoven, The Netherlands. [3]These authors contributed equally: David van Driel, Guanzhong Wang, and Tom Dvir. ✉e-mail: tom.dvir@gmail.com

This couples the even-occupation levels of the semiconductor, whereby they become ABSs with an induced pairing gap $\Gamma$[15–20]. While this exchange of electrons does not conserve charge, the parity of an ABS remains well-defined: even or odd. In the case of even parity, the ABS is in a singlet state which, within the atomic limit[15], is of the form:

$$|S\rangle = u|0\rangle - v|2\rangle \qquad (1)$$

where $|0\rangle$ denotes the state in which the ABS is unoccupied and $|2\rangle$ the state in which it is occupied by two electrons. $u$ and $v$ are the relevant BCS coefficients[15,21]. The odd-parity manifold consists of a doublet of two states, $|\downarrow\rangle$ and $|\uparrow\rangle$, which are degenerate in the absence of an external magnetic field $B$. The energies of the even singlet and odd doublet states, as well as $u$ and $v$, depend on $\mu_H$, the energy of the uncoupled quantum level with respect to the superconductor Fermi energy. Both are shown schematically in Fig. 1a, b for Zeeman energy $E_Z = 3\Gamma$. A finite Zeeman energy $E_Z$ splits the doublet states in energy, while the singlet does not disperse (Fig. 1c). An ABS can be excited from its ground state to an excited state of opposite parity by receiving or ejecting a single electron from a nearby reservoir. This parity-changing process requires an energy $\xi$, the energy difference between the ground and excited states, as indicated in Fig. 1a, c. These excitation energies are detected as conductance resonances in conventional tunneling spectroscopy measurements[8,22–26]. The ABS resonances split with an applied magnetic field when the ground state is even and disperse to higher energy with an odd ground state[23]. These ABS excitations are believed to be spin-polarized as they arise from transitions between spinless and spin-polarized many-body states. Spin polarization weakens in the presence of spin-orbit coupling,

where ABSs become admixtures of both spins and different orbital levels[27]. A pseudo-spin replaces spin as the quantum number defining the doublet states, and complete spin-polarization along the applied field direction is no longer expected[28]. Measurement of the spin polarization of ABS excitations may thus reveal the presence of spin-orbit coupling in hybrid systems. So far, spin-polarized spectroscopy of comparable Yu-Shiba-Rusinov states[29–31] has indeed revealed signatures of finite spin polarization on ferromagnetic adatoms[32–35]. It was further argued that the observation of fully spin-polarized zero-energy edge modes in ferromagnetic chains is a strong signature of a topological phase[36,37].

In this work, we measure the spin-polarized excitation spectrum of a hybrid InSb nanowire hosting an ABS. This is done using a three-terminal setup consisting of a grounded superconductor–semiconductor hybrid tunnel-coupled on one side to a spin-polarized quantum dot (QD) and on the other side to a conventional tunnel junction. At low magnetic fields, we show complete spin polarization of the ABS, which reverses with increasing fields. At even higher fields, we observe a persistent, spin-polarized zero-bias peak. Furthermore, we show that the complete spin polarization of the ABS is responsible for a transport blockade that can be lifted by coupling the ABS to a non-polarized electron reservoir. We refer readers to a simultaneous submission by Danilenko et al.[38] for another report reaching complementary conclusions.

## Results and discussion
### Device fabrication and set-up
The fabrication of the reported device follows ref. 13. An InSb nanowire was placed on an array of bottom gates which are separated from the nanowire by a thin bilayer of atomic-layer-deposited (ALD) $Al_2O_3$ and

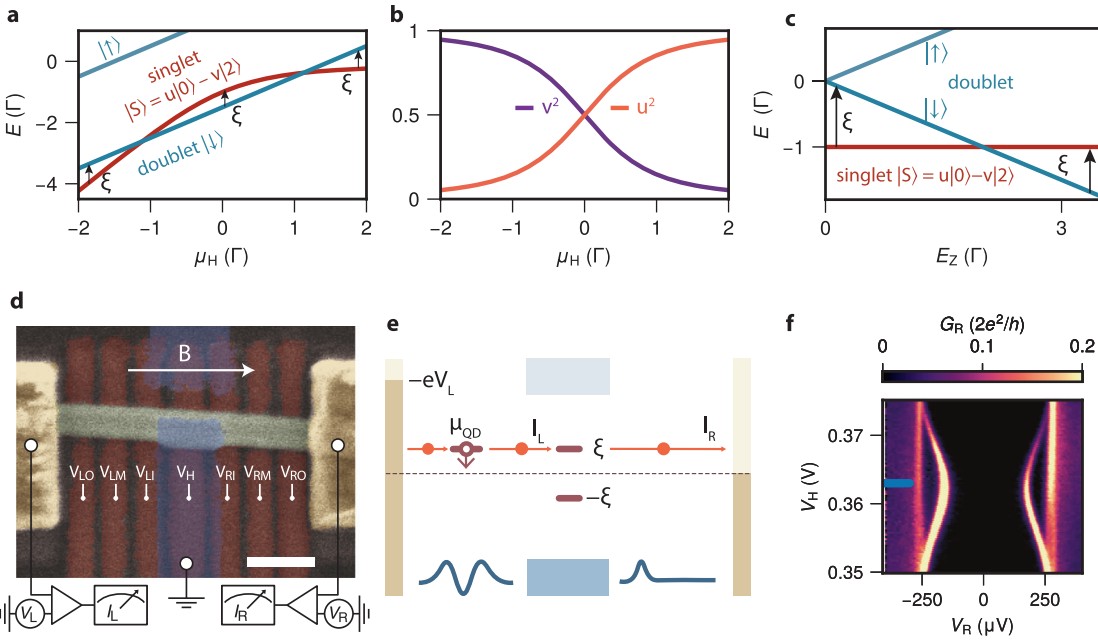

**Fig. 1 | Tunnel spectroscopy in a three-terminal InSb-Al nanowire. a** Energy diagram showing the evolution of the many-body Andreev bound state (ABS) spectrum with electrochemical potential $\mu_H$ for $E_Z = 3\Gamma$. The arrows illustrate the parity-changing transition energies $\xi$ from the ground state to the first excited state. **b** The dependence of $u^2$ and $v^2$ on $\mu_H$ under the same conditions as a. **c** Energy diagram showing the ABS spectrum with the applied magnetic field creating Zeeman splitting $E_Z$, at $\mu_H = 0$. All calculations in a–c are made in the atomic limit approximation with zero charging energy[15]. **d** False-colored SEM image showing the device studied throughout the paper. Normal leads are yellow, bottom gates red, the InSb nanowire green and the grounded Al shell is blue. The normal contacts can be biased independently with respect to the grounded Al, the current is

measured left and right simultaneously. Scale bar is 200 nm. **e** Schematic diagram of electron transport using a quantum dot (QD) as a spin filter. Finger gates define a QD and a tunnel junction in the nanowire (blue lines at the bottom sketch their potential profiles). Lead electrochemical potentials are indicated by the dark yellow rectangles. The left lead is biased at a voltage $V_L$ with respect to the grounded superconductor (blue). The QD states are at a gate-tunable energy $\mu_{QD}$. ABS excitation energies are shown as brown, horizontal lines. The blue rectangles indicate the Al quasiparticle continuum. The potential landscape created by the gates is shown schematically by the blue lines at the bottom of the panel. **f** Tunneling spectroscopy result $G_R$ of the investigated ABS for varying gate voltage $V_H$ for $V_{RO} = 500$ mV.

HfO$_2$ dielectric of ~10 nm each. A thin Al segment of length roughly 200 nm was evaporated using a shadow-wall lithography technique[39,40]. Normal Cr/Au contacts were fabricated on both sides of the device (more details on substrate fabrication can be found in ref. [41]). Figure 1d shows the device along with its gates. Throughout the experiment, we keep the middle superconductor grounded. Both left and right leads are voltage-biased independently with biases $V_L$ and $V_R$, respectively. Similarly, the currents on the left and right leads ($I_L$ and $I_R$, respectively) are measured simultaneously. The full circuit is shown in Supplementary Fig. 1 and is discussed in the methods.

Figure 1e sketches the energy diagram of electron transport in the device. The left three gates define a QD in the InSb segment above them, whose electrochemical potential $\mu_{QD}$ is controlled linearly by $V_{LM}$. The three gates on the right side define a conventional tunnel junction. Using this circuit, the hybrid can be probed in two ways. When the QD is off-resonance and the left side does not participate in transport, conventional tunnel spectroscopy can be performed from the right. The resulting $G_R = dI_R/dV_R$ is shown in Fig. 1f, revealing a discrete sub-gap ABS, described by the intermediate regime of superconductor–semiconductor coupling[7]. The energy of the ABS disperses with $V_H$, which linearly relates to $\mu_H$. All further results are obtained at $V_H = 363$ mV as indicated by the blue line unless otherwise specified. This places the ABS in the vicinity of its energy minimum. The second way of examining the ABS is performing QD spectroscopy from the left, by applying a fixed $V_L$ and varying the probed energy by scanning $\mu_{QD}$. Setting $-eV_L > \mu_{QD} = |\xi| > 0$ injects electrons into the ABS, while $-eV_L < \mu_{QD} = -|\xi| < 0$ extracts electrons from the ABS. In the presence of a Zeeman field, the QD charge transitions become spin-polarized when $2E_Z > e|V_L|$, allowing only spins of one type to tunnel across it[42]. As a result, the QD is operated as a bipolar spin filter with a finite energy resolution (see Methods and Supplementary Fig. 2). We use ↑, ↓ to represent the two spin polarities and label the QD chemical potentials $\mu_{QD\uparrow,\downarrow}$ to distinguish between them where necessary.

We note that our spin probe consists of only a single quantum state which becomes fully spin-polarized under the presence of a large Zeeman field. This is different from conventional spin-polarized tunneling in scanning tunneling microscopy experiments, where an exemplary Fe tip achieves 40–45% polarization[43]. In a more recent study using a YSR-state on a STM tip as a spin probe, the filtering mechanism using a single quantum state is comparable to that reported in this paper[35].

## Zeeman-driven singlet–doublet transitions

Figure 2a shows tunneling spectroscopy of the particular ABS shown in Fig. 1f for varying $B$. The ABS conductance peak at $|V_R| \approx 200\,\mu V$ Zeeman-splits into two resonances, one moving to higher and the other to lower energies. At $B \approx 300$ mT, the low-energy states cross at zero energy. This crossing has been identified earlier[23] as a singlet–doublet transition, where the ground state of the hybrid becomes the odd-parity $|\downarrow\rangle$ state. Next, we perform QD spectroscopy by measuring $I_L$ as a function of $V_{LM}$ with fixed $V_L = \pm 400\,\mu V$. For each spin, the spectrum is obtained by converting $V_{LM}$ to $\mu_{QD}$ and then combining the two bias polarities (see Methods and Supplementary Fig. 2). Figure 2b and c show the resulting QD spectroscopy for varying $B$. The QD functions as a ↓-filter (panel b) or ↑-filter (panel c) for $B > 100$ mT (blue dashed lines), where the QD Zeeman energy splitting exceeds $|eV_L|$ and only one spin is available for transport. The white triangle in Fig. 2b indicates missing data for $V_L > 0$. Finite current within this triangle arises due to peak-broadening in $V_L < 0$ data (see Supplementary Fig. 4 for details).

The peaks observed in QD spectroscopy are also visible in tunneling spectroscopy (see Supplementary Fig. 5 for comparison), although only one branch of the particle-hole-symmetric peaks in $G_R$

remains for each QD spin. The down-filtered ABS feature (panel b) disperses with a negative slope. We understand this by examining the energy diagram in Fig. 1c. For $|B| < 300$ mT, the non-dispersing singlet is the ground state and the first excited state is $|\downarrow\rangle$. Probing the ABS at positive energies injects spin-down electrons, which excites the singlet to $|\downarrow\rangle$ as illustrated in the upper part of Fig. 2f. Hence, these peaks in electron transport move down with increasing $B$. At $B \approx 300$ mT, the ABS undergoes a quantum phase transition, after which the ground state is $|\downarrow\rangle$. To transition from $|\downarrow\rangle$ to $|S\rangle$ without participation of spin-up electrons, a down-polarized electron must be first removed from the ABS, as shown in the upper part of Fig. 2g. Thus, the peak in current is found only for $\mu_{QD} < 0$. For $B < 0.6$ T, the ↑-filter data in panel c mirrors that of panel b. This symmetry is understood by comparing the lower parts of Fig. 2f, g to the respective upper ones. When the ABS allows injecting spin-down electrons, it also allows removing spin-up ones, due to $|S\rangle$ being a superposition of empty and doubly-occupied states.

Thus far, we have focused on the excitation of the ABS. To complete a transport cycle, the ABS must also be able to relax back to the ground state. This can be done by either emitting or accepting electrons from the right lead through the tunnel junction. As a consequence, finite QD-current $I_L$ is generally accompanied by finite $I_R$ at the corresponding energy and field, as shown in Fig. 2d, e. We observe the same sub-gap features as in panels b,c confirming that these are extended ABSs that couple to both normal leads. Upon crossing the singlet–doublet transition, the ABS relaxation requires an opposite direction of electron flow, giving rise to the sign switching of $I_R$ at $B = 0.3$ T[8,10–13,44]. The precise sign of $I_R$ depends on $\mu_H$ as discussed in more detail in below.

We further quantify the spin-polarization of the ABS in Fig. 2h. We calculate the spin polarization $P = (I^\downarrow - I^\uparrow)/(I^\uparrow + I^\downarrow)$, where $I^\uparrow$, $I^\downarrow$ indicates the current measured using the down- or up-polarized configuration, using the data from Fig. 2b, c (see Supplementary Fig. 5 for details and Methods for the definition of $I^\uparrow$, $I^\downarrow$). Before the singlet–doublet transition, we see a fully spin-polarized ABS with $P = \pm 1$. The spin polarization reverses immediately after the transition. At the singlet–doublet transition, we observe a zero-bias peak with $|P| < 1$ as both spins can participate in transport. The non-vanishing calculated $P$ may be due to microscopic details in QD transport (see Methods).

We emphasize that we report on complete spin polarization, i.e., the rate of exciting the ABS to the $|\downarrow\rangle$ state by injecting an up-polarized electron is below noise level, which is ~1 pA. This indicates that no noticeable spin rotation occurs during tunneling between the QD and ABS and their spin must be co-linear. Spin rotation is predicted to arise in the presence of strong spin-orbit coupling, which was observed in a similar setup in our previous work[13] (See Supplementary Fig. 7 for spin-polarized measurements conducted with that device, showing incomplete spin-polarization). We attribute the absence of spin rotation for this particular ABS to the large level spacing in both the QD (Supplementary Fig. 3) and the ABS (Fig. 1d) preventing efficient spin-mixing between different orbital states[27]. See Supplementary Fig. 9 for a large gate range measurement of QD levels splitting in field.

The presence of spin-flip tunneling, induced by the spin-orbit interaction, could also result in the lifting of the observed blockade[45]. Previously, such spin-flip tunneling in InAs-based double quantum dots was modulated by controlling the barrier separating the QDs[46]. In Supplementary Fig. 10 we present spin-polarized spectroscopy taken with a more positive tunnel gate value $V_{LI}$, showing a small lifting of the spin blockade. Since we do not expect the change in the tunnel gate voltage to significantly affect the QD or the ABS levels, we interpret the partial lifting of the spin blockade as resulting from spin-flip tunneling.

At higher fields ($B > 0.6$ T), we observe another low-energy ABS in tunneling spectroscopy. Above $B \approx 0.75$ T, this ABS sticks to zero energy and is completely down-polarized (Fig. 2h). While a persistent,

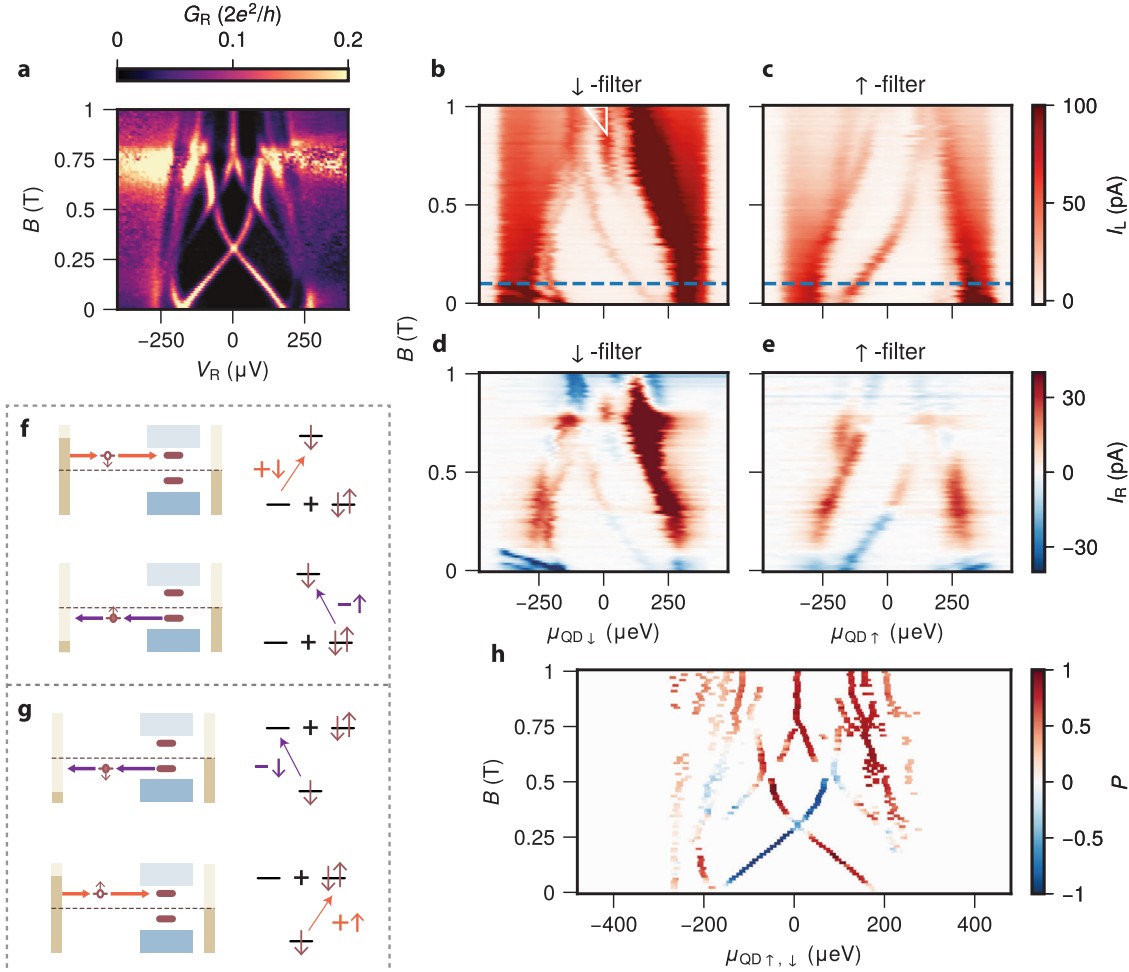

**Fig. 2 | Spin-polarized quantum dot spectroscopy of an Andreev bound state across the Zeeman-driven singlet–doublet transition. a** Tunneling spectroscopy of the hybrid for $B$ applied along the nanowire axis for $V_{RO} = 500$ mV and $V_H = 363$ mV. **b, c** $I_L$ vs $\mu_{QD}$ and $B$ using the quantum dot (QD) as a ↓-filter (**b**) and ↑-filter (**c**). The blue dashed line at $B = 100$ mT indicates the field above which the QD becomes a spin filter. **d, e** $I_R$ vs $\mu_{QD}$ and $B$ using the QD as a ↓-filter (**d**) and ↑-filter (**e**). **f** Schematic energy diagram of spin-polarized excitations for the singlet ground state Andreev bound state (ABS). A down spin can tunnel into the ABS (upper) or an up spin can tunnel out (lower). **g** Schematic energy diagram of spin-polarized excitations for the doublet ground state ABS. A down spin can tunnel out of the ABS (upper) or an up spin can tunnel into it (lower). **h** The spin-polarization $P = (I^\downarrow - I^\uparrow)/(I^\uparrow + I^\downarrow)$ calculated from **b, c** at ABS energies found from **a**[62].

spin-polarized zero-bias peak is a predicted signature of Majorana bound states[47–49], the short length of our hybrid section excludes this interpretation[50]. The interplay of spin-orbit coupling and confinement is a known mechanism for the formation of persistent zero-bias conductance peaks in QDs coupled to superconductors, given precise tuning of the QD chemical potential[28]. Such states develop a spin texture that has a global vanishing magnetic moment but is locally spin-polarized[51]. Our observation of a spin-polarized, persistent zero-bias peak is consistent with this interpretation. We emphasize that this state is fine-tuned using $V_H$ (see Supplementary Fig. 8 where for a different value of $V_H$ the zero-bias peak does not persist over a large range of $B$) and further study is required to fully characterize such states.

**Gate-driven singlet–doublet transition**

The singlet–doublet phase transition reported above can also occur upon gate-tuning the electrochemical potential of the ABS, as illustrated in Fig. 1a. Figure 3a shows tunneling spectroscopy for varying $V_H$ at $B = 350$ mT. The ABS crosses zero at $V_H = 0.357$ V and $V_H = 0.366$ V. Between these two crossings, the ground state of the ABS is $|\downarrow\rangle$. At higher and lower values of $V_H$, the ground state is the singlet $|S\rangle$. This is observed in the spin-polarized spectroscopy shown in Fig. 3b,c. $|S\rangle$ can only be excited to $|\downarrow\rangle$ when spin-down electrons

tunnel into the hybrid or when spin-up electrons tunnel out. The doublet ground state shows the opposite: The ↓-filter peak in current is found only for $\mu_{QD} < 0$ and the ↑-filter peak only for $\mu_{QD} > 0$.

The non-local relaxation current $I_R$ (panels d and e) shows three alternations between positive and negative currents, for both spin polarizations. At lower values of $V_H$, $|u|^2 \gg |v|^2$ and $|S\rangle \approx |0\rangle$. Therefore, the dominant relaxation mechanism from the excited $|\downarrow\rangle$ state to the ground state $|S\rangle$ is an electron tunneling out of the ABS to the right lead, giving rise to positive $I_R$. At high $V_H$, $|S\rangle \approx |2\rangle$ and relaxation entails electrons tunneling into the ABS and thus $I_R < 0$. At the two singlet–doublet transitions (dotted lines in panel d), the current sign reverses for the same reason discussed in Fig. 2d, e. At $V_H = 361$ mV (dashed line in panel d), $\mu_H$ crosses zero and the effective charge of the ABS, $|u|^2 - |v|^2$, switches sign. The non-local current also reverses direction, an effect investigated in detail in literature[8,10–13,44].

Figure 3g shows the corresponding spin polarization that was computed likewise to Fig. 2h. Transport at positive $\mu_{QD}$ is seen to be fully down-polarized in the singlet ground state. Likewise, states are up-polarized for negative $\mu_{QD}$. Polarization appears as incomplete in the $|\downarrow\rangle$ ground state. This is a measurement artifact due to using QDs as spin filters. The up-, and down-filter transport peaks have finite broadening due to temperature and coupling to the leads. For ABSs close to zero, this can result in peaks at negative and positive $\mu_{QD}$

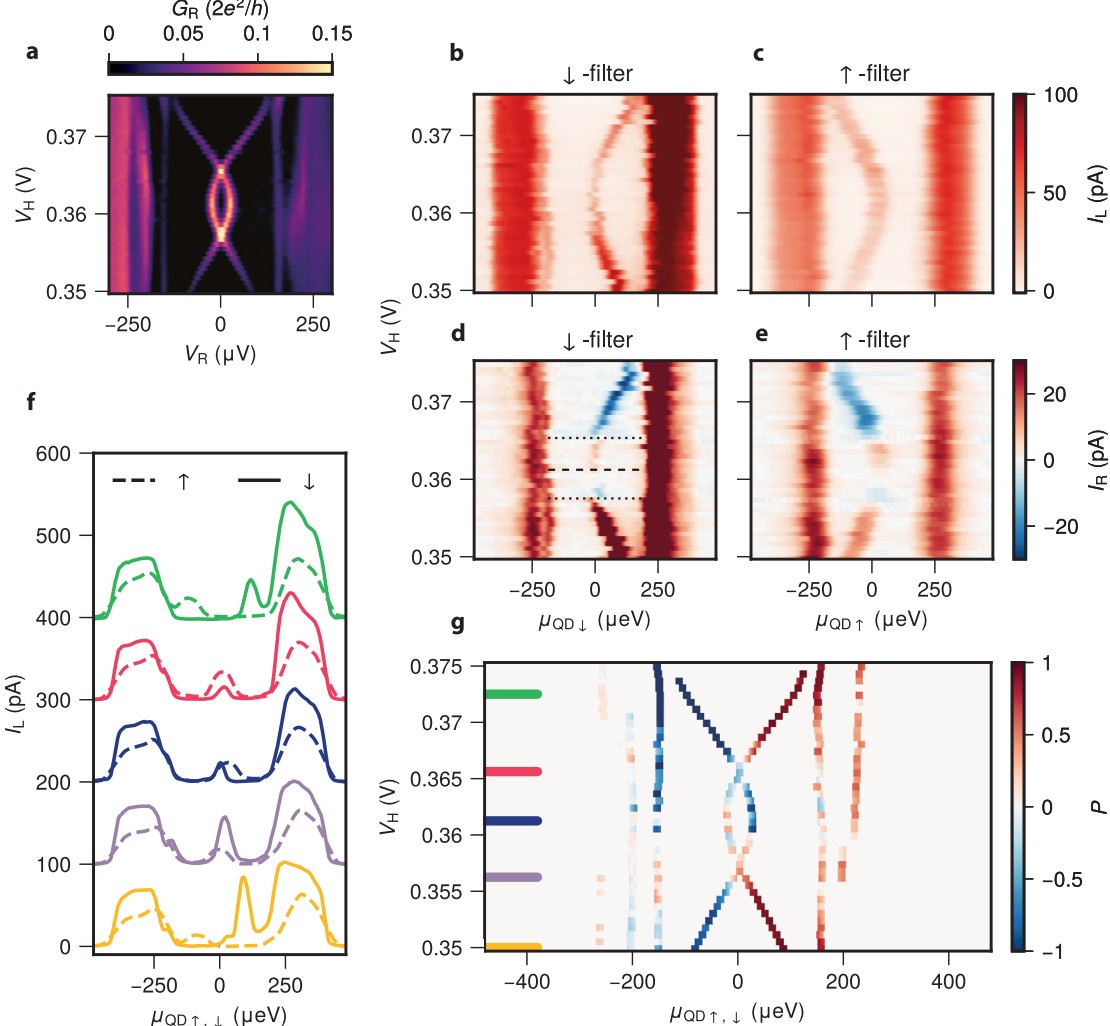

**Fig. 3 | Spin-polarized quantum dot spectroscopy of an Andreev Bound State during the gate-driven singlet–doublet transition. a** Tunneling spectroscopy of the hybrid for varying $V_H$. The external magnetic field is fixed at $B = 350$ mT and $V_{RO} = 500$ mV. **b, c** $I_L$ vs $\mu_{QD}$ and $V_H$ using the quantum dot (QD) as a ↓-filter (**b**) and ↑-filter (**c**). **d, e** $I_R$ vs $\mu_{QD}$ and $V_H$ using the QD as a ↓-filter (**d**) and ↑-filter (**e**). The non-local current changes sign three times, twice at the singlet-doublet transitions (black dotted lines) and once at the Andreev bound state energy minimum (black dashed line). **f** Linecuts of **b, c** at values of $V_H$ indicated by the lines in g. Each pair of traces is offset by 100 pA for readability. **g** The spin-polarization $P = (I^↓ - I^↑)/(I^↑ + I^↓)$ found from panels **b, c**.

overlapping, as seen in the dark blue linecut of Fig. 3f, giving imperfect polarization.

A similar problem occurs for states close to the gap edge. We see that the higher energy ABS always appears down-polarized at positive $\mu_{QD}$ and up-polarized at negative $\mu_{QD}$. Close to the gap edge, broadening results in tunneling into the Al density of states, in addition to the ABS. The inability to distinguish between tunneling into the metallic and semiconductor density of states makes the interpretation of spin polarization at higher energies unreliable.

**The Andreev bound state relaxation mechanism**

To emphasize the role of the right lead in relaxing the ABS, we show the effect of decoupling it from the hybrid. Figure 4a shows $G_R$ for varying $V_R$ and $V_{RO}$ at $B = 200$ mT, for which the ground state is singlet. Lowering the gate voltage $V_{RO}$ gradually decouples the hybrid from the right normal lead, evident in the decay of $G_R$. For $V_{RO} < 0.28$ V, the right junction is fully pinched off. Figure 4b shows the corresponding QD spectroscopy using the ↑-filter. The transport at energies exceeding the superconducting gap is virtually unaffected by the pinching-off of the normal lead. Strikingly, transport between the QD and the ABS is completely blocked for $V_{RO} < 0.28$ V. In addition, the non-local

transport is suppressed at all energies once the right lead is pinched off (Fig. 4c). To understand this blockade, we first consider the full transport cycle. Figure 4d, e illustrate the ABS excitation and relaxation processes. The ABS is excited from the singlet ground state to the $|↓\rangle$ excited state by ejecting an electron into the ↑-filter QD (panel d). The spin-down electron in $|↓\rangle$, however, cannot tunnel out into the spin-up left QD again. For the ABS to transition back into the ground state and restart the transport cycle, the right side has to participate in its relaxation. This is done by either receiving an up electron from the non-spin-polarized right lead or, interestingly, by emitting a down electron to the right lead (panel e). As discussed above, the singlet state is a superposition $|0\rangle$ and $|2\rangle$, coupled via Andreev reflection. Decoupling the non-polarized right lead from the hybrid removes the only source of relaxation of the ABS, resulting in the transport blockade seen in panel b. We emphasize that this differs from the spin-filtering effects discussed in previous sections, where incompatible spin states between the QD and the ABS prevent the excitation of the ABS. It is instead similar to the so-called Andreev blockade[52] where the excitation is allowed, but the relaxation process is suppressed. Our current noise level of ~1 pA gives a lower bound for the spin-relaxation time of ~100 ns. ABS parity lifetimes of 10 μs or slower have been

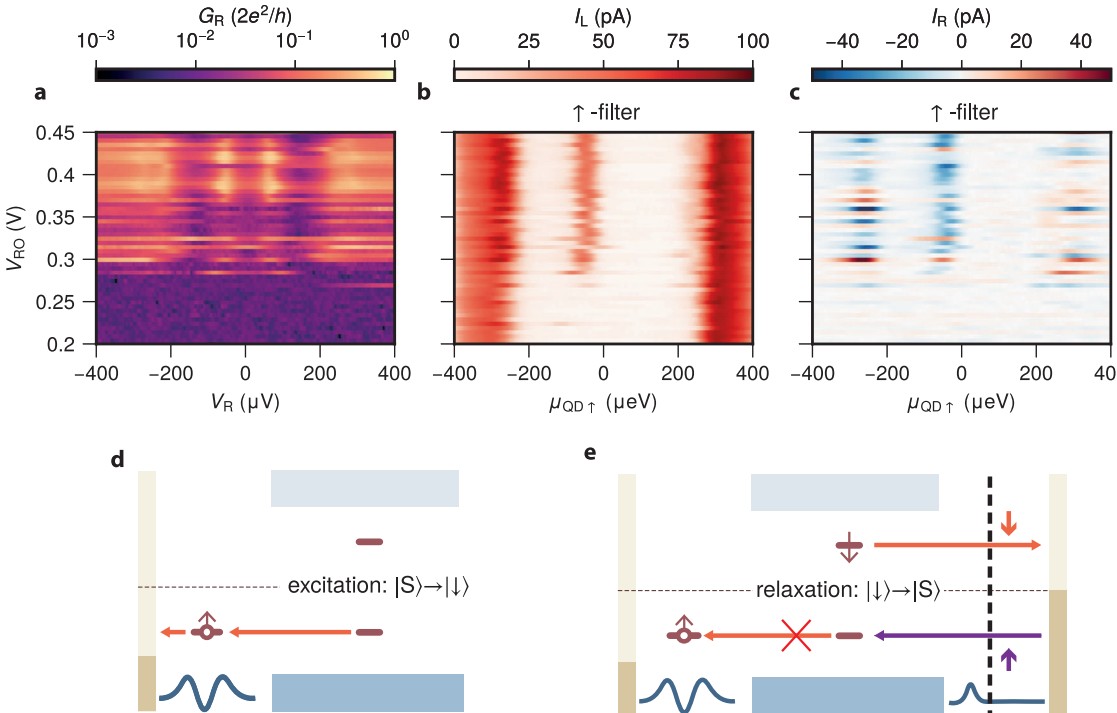

**Fig. 4 | Blocking the Andreev bound state excited state relaxation. a** Tunneling-spectroscopy $G_R$ as a function of $V_R$ and $V_{RO}$ at $B = 200$ mT for $V_H = 363$ mV. Transport through the right tunnel junction is pinched off for $V_{RO} \lesssim 0.28$ V. **b** Quantum dot (QD) spectroscopy $I_L$ for the ↑-filter for varying $V_{RO}$. Transport through the Andreev bound state (ABS) stops when the tunnel junction is pinched off. **c** QD-spectroscopy $I_R$ for the up-polarized QD level for varying $V_{RO}$. All non-local transport stops when the tunnel junction is pinched off. **d** Schematic illustration of ABS excitation under the spin and bias setting in **b**, **c**. The ABS can only be excited from $|S\rangle$ to $|\downarrow\rangle$ via ejecting an electron into the QD. **e** Schematic illustration of the ABS relaxation process. Transitioning from $|\downarrow\rangle$ back to $|S\rangle$ is only possible via electron exchange with the right lead and becomes blocked when parts to the right of the vertical black dashed line are pinched off.

observed[53], which is significantly slower than we can resolve. We interpret this to mean that there are no mechanisms in our system that can relax the ABS in < 100 ns.

In conclusion, we use a spin-polarized QD to characterize the spin-polarization of an ABS. We show complete spin polarization of the ABS excitation process, signaling the absence of significant spin-orbit coupling in the measured regime. We further observe the reversal of the ABS spin and charge when driving it through the singlet–doublet transition using applied magnetic field or gate voltage. Furthermore, we note the appearance of a spin-polarized zero-bias peak at higher magnetic fields. This work establishes the use of spin-polarized QDs as a spectroscopic tool allowing the study of the spin degree of freedom. This can be utilized in the future to study topological superconductors, where a reversal of the bulk spin-polarization is expected when the system is tuned to the topological regime[54]. The predicted spin polarization of the arising Majorana zero modes can also be observed in this way[55].

## Methods

### Device set-up
Supplementary Fig. 1 shows a sketch of the device with the electrical circuit used for the experiment. The Al was evaporated at two angles relative to the nanowire: 5 nm at 45° and 4.5 nm at 15°, forming a superconductor–semiconductor hybrid underneath. The Al shell is kept grounded during the experiment. Voltage bias is applied on either the left lead ($V_L$) or the right lead ($V_R$) while keeping the other lead grounded. The currents on both leads ($I_L$ and $I_R$ on the left and right leads, respectively) are measured simultaneously. A small RMS AC amplitude $V_R^{AC} = 4\,\mu$eV is applied on top of $V_R$ for the tunneling spectroscopy measurements. All measurements are performed in a dilution refrigerator with a base temperature of 30 mK. The magnetic field is applied along the wire length. The quantum dot on the left was formed

by creating two tunnel junctions using $V_{LI}$ and $V_{LO}$. Its electrochemical potential is controlled by $V_{LM}$. The tunnel junction on the right is formed by creating a single tunnel junction with $V_{RI}$. The electrochemical potential of the hybrid section is set using $V_H$. Pinching off the tunnel junction as explained in Fig. 4 was done by changing the value of $V_{RO}$.

### Quantum dot spin filter
The QD is characterized by measuring the current on the left lead ($I_L$) as a function of $V_L$ and $V_{LM}$. Supplementary Fig. 2a shows a single Coulomb diamond with an orbital level spacing of $\delta = 3.5$ meV, much larger than the superconducting gap and Zeeman energy used throughout the experiment. The large level spacing allows us to treat the QD as a single orbital near the Fermi energy, which is occupied by zero, one, or two electrons, as indicated in the figure (see measurement with an extended range of $V_L$ and $V_{LM}$ in Supplementary Fig. 3). From this measurement we find a lever-arm for $V_{LM}$ of $\alpha = 0.4$. The current $I_L$ is completely suppressed for $|eV_L| < 170\,\mu$eV, indicating a hard gap and no local Andreev reflection for these gate settings[56]. The current on the right lead, $I_R$, which was measured simultaneously, shows similar features (Supplementary Fig. 2b). To analyze the data, we first convert $V_{LM}$ to the electrochemical potential of the QD ($\mu_{QD}$) at a fixed negative or positive $V_L = \pm 400\,\mu$V (orange and purple lines in panel a, respectively). For a given negative bias $-eV_L > 0$ (see schematic drawing in Supplementary Fig. 2c), the Fermi energy of the lead lies above that of the hybrid segment. Hence, electrons are injected into the hybrid when $\mu_{QD}$ is within the transport window: $-eV_L > \mu_{QD} > 0$. We then convert the value of $V_{LM}$ to $\mu_{QD}$ through $\mu_{QD} = -(\alpha e(V_{LM} - V_{LM}^0) + eV_L)$, where $V_{LM}^0$ is the gate voltage at which the dot level is aligned with the applied bias. Note that $\mu_{QD} = -eV_L$ when the QD is aligned with the bias edge $V_{LM} = V_{LM}^0$. See Supplementary Fig. 4 for a comparison of raw and processed data. In Supplementary Fig. 2d we plot $|I_L|$ vs $\mu_{QD}$. The

current shows two peaks at ~ 270 $\mu$eV and ~ 170 $\mu$eV, which we interpret as the bulk superconducting gap and the ABS energy, respectively. Similarly, for positive bias, the Fermi energy of the lead lies below that of the hybrid segment and electrons tunnel out of the hybrid segment when $\mu_{QD}$ is within the transport window (see schematic drawing in Supplementary Fig. 2e). In Supplementary Fig. 2f, we plot $|I_L|$ vs $\mu_{QD}$ and see features that are symmetric to those shown in panel d. We use the positive-bias data for $\mu_{QD} < 0$ and negative-bias data for $\mu_{QD} > 0$. Juxtaposing the two halves yields the full spectrum in Supplementary Fig. 2g. The spectrum obtained in this way is in qualitative agreement with the superimposed tunneling spectroscopy results at the same $V_H$. Therefore, measuring the current through the QD enables us to obtain the ABS spectrum[57,58]. Next, we apply an external magnetic field to polarize the QD excitations. The even-to-odd QD charge transition ($V_{LM} \approx 369$ mV) now involves only the addition and removal of electrons with spin ↓, while the odd-to-even transition ($V_{LM} \approx 379$ mV) involves the addition and removal of electrons with spin ↑[27,59]. As a result, the QD becomes a spin filter, allowing spin-polarized spectroscopy[42,60].

### Analysis of spin-polarization

To compute $P$ for Figs. 2h and 3h, we first find peaks in tunneling spectroscopy using a standard peak-finding procedure provided by the SciPy python package[61]. The peak energies found from Fig. 2a are indicated by white dots in Supplementary Fig. 5a. Supplementary Fig. 5b, c shows these peak energies, overlaid on the ↓ and ↑-filter data of Fig. 2. For each of these energies, we finally calculate $P = (I^{\downarrow} - I^{\uparrow})/(I^{\uparrow} + I^{\downarrow})$. The data processing is done similarly for Fig. 3h.

This procedure correctly produces $P = \pm 1$ when spin-polarization is complete, i.e., $I^{\uparrow} = 0$ while $I^{\downarrow} \neq 0$ at a given $\mu_{QD}$ or vice versa. When polarization is incomplete, however, our calculated $P$ may differ quantitatively from the true spin composition of the ABS. For example, measuring an entirely non-polarized $P = 0$ requires $I^{\uparrow} = I^{\downarrow}$. In practice, since the two spin filters are different QD charge degeneracies and differ in gate voltage, $I^{\uparrow}$ is often different from $I^{\downarrow}$ even at zero field due to electrostatic effects on the tunnel rate, leading to finite calculated $|P|$. This is indeed the case in Fig. 2b, c. From Fig. 2h, we see $|P| < 1$ at $B \approx 300$ mT. Because the ABS has $P = \pm 1$ before and directly after the singlet–doublet transition, we conclude that its spin polarization, if any, must be weak.

## Data availability

The data generated in this study, as well as the code used to analyse and plot it have been deposited in a Zenodo repository that can be accessed freely at: https://doi.org/10.5281/zenodo.7220682.

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

## Acknowledgements

This work has been supported by the Dutch Organization for Scientific Research (NWO) and Microsoft Corporation Station Q. We wish to acknowledge useful discussions with Michael Wimmer, and Alisa Dani-lenko and support from Gijs de Lange.

## Author contributions

D.v.D., G.W., and T.D. contributed equally to this work. D.v.D., G.W., A.B., N.v.L., F.Z., and G.P.M. fabricated the devices. D.v.D., G.W., T.D., D.X., and F.Z. performed the electrical measurements. D.v.D., T.D., and G.W. designed the experiment and analyzed the data. D.v.D., T.D., and L.P.K. prepared the manuscript with input from all authors. T.D. and L.P.K. supervised the project. S.G., G.B., and E.P.A.M.B. performed InSb nanowire growth.

## Competing interests

The authors declare no competing interests.
