## [Peer Review File · Nature Communications]

Reviewers' Comments:

Reviewer #2:

Remarks to the Author:

The manuscript reports a detailed study of spectroscopy of Andreev bound states (ABS) in a semiconductor nanowire with strong spin-orbit coupling (SOC). These systems are highly interesting in the context of the search for topological states. The main claim is the determination of the spin state of the ABS. I would recommend publication in Nature Communications if the authors can address the following point:

- can the authors give a full characterization of the spin states of left dot (the spin filter)? What is the g factor? What is the SOC strength? The authors claim that at 100 mT the Zeeman splitting is larger than V_L , which is larger than the induced gap. In contrast, the Zeeman shift of the ABS in the center dot is clearly much less than the gap at 100 mT. Also, the claim of full spin polarisation depends on the nature of the states in the left dot, which is subject to the same SOC as the center section with the ABS.

minor points:

- Ref. 9 is missing bibliographic data
- ED8 was probably measured with only one of the gates on the left dot operated to get a simple tunnel junction. This should be mentioned explicitly in the caption, otherwise it is confusing in the context of the other experiments.

Reviewer #3:

Remarks to the Author:

In this paper, the authors implement spin-filtering experiments of the Al-InSb nanowires. The proximitized region of the InSb nanowire is studied by tunnel spectroscopy. They reproduce the well-known singlet-doublet transition invoked by the Zeeman splitting reported in similar nanowire experiments in the literature. An important progress is that they use the quantum dot adjacent to the proximitized region as a spin-filter probe under the magnetic field. Thanks to the QD spin probe, they succeed in the evaluation of the spin polarization of the excitation from the singlet or doublet ABSs. Consequently, they find the perfect spin polarization despite the spin-orbit interactions in the nanowire. In addition, they address relaxation from the excited state by switching the nonlocal probe. Although the spin-filter experiments have already been reported in the STM experiments as cited in the manuscript, their data quality seems much better than the previous STM experiments. In addition, the blockade of the relaxation of the ABS seems important to establish ABS physics. Their device structure is expected to be a platform for detecting and controlling the Majorana Fermions, which is currently attracting much attention, and the authors mention the relevance of the Majorana Fermions in their discussion of the magnetic field dependence.

The paper provides useful and valuable information for those who study the physics of superconductor-semiconductor junctions. However, I have several questions and comments on the results and discussion before I recommend the publication.

I have the following comments:

1. An electrical circuit in Fig. ED1 should appear in the main manuscript instead of Fig. 1d otherwise it seems difficult to understand what they measure.
2. They discuss the possible spin-flip process which can degrade the spin-filtering. The spin-flip can happen because the effective field direction of the SOI is perpendicular to the external field. They propose that the energy level spacing is large (compared to what?) enough to suppress the spin-flip, resulting in the perfect spin polarization. I cannot understand why this large spacing can

prevent spin-flip tunneling. Even if the large energy splitting in the QD and the ABS, the spin-flip seems to be invoked by the SOI between the up-spin state in QD and the down-spin state of the ABS. A more detailed discussion is necessary.

3. Related to the above question, I want to know the differences between this device and their previous device [9]. Why does the spin-flip tunneling happen in the previous device?

4. I recommend representing their gate conditions (at least V_H and V_{RO}) in the captions for all the figures (or directly in the figures).

5. I could not find the discussion of Figs. 2d and e in the section of "Zeeman-driven singlet-doublet transitions". Are they necessary in the main manuscript? For the discussion of relaxation, Figure 4 is sufficient for me.

6. They discuss the zero energy state as a quasi-Majorana state. However, I am very confused about the definition of the "quasi-" Majorana state. Do they want to discuss the interference of the two Majorana at both edges? If so, they need to represent the interference pattern by changing the magnetic field or similar results. Otherwise, they should not use the "Majorana" state as their interpretation.

7. They should cite the original theory papers of the Yu Shiba Rusinov states.

8. I expect that the sum of the spin-polarization of the excited ABS (for example in Fig. 3g) should be zero. They have better exhibit the analysis results and discuss the reason if the sum at a certain region is not zero.

9. They nicely address the relaxation of the excited ABS by controlling the nonlocal current I_R . This is impressive to me. In their results, there seems to be no leakage current, and the relaxation is completely blocked when V_{RO} pinches off the right nanowire region. I want to know more details about why the excited ABS are not leaked to the ground state. I think that the quasiparticle poisoning time decides the blockade time. Does this result indicate that the poisoning time is sufficiently long?

10. As they also mention, the spin-filtering experiments of the YSR or ABS have already been reported in the STM measurements. However, the previous results do not indicate the perfect spin-polarization if I understand correctly. Then if possible, please discuss why they can obtain the perfect polarization and what is the most critical ingredient or advantage in their device or setup.

11. I think that the manuscript title is too simple. This title may mislead that this study is the first experimental work about the spin-filtering measurement of the ABS. I recommend changing their title (for example they need to clarify they use the superconductor-semiconductor hybrid devices).

Reviewer #2 (Remarks to the Author):

The manuscript reports a detailed study of spectroscopy of Andreev bound states (ABS) in a semiconductor nanowire with strong spin-orbit coupling (SOC). These systems are highly interesting in the context of the search for topological states. The main claim is the determination of the spin state of the ABS. I would recommend publication in Nature Communications if the authors can address the following point:

We thank the reviewer for the critical review and the positive assessment of the manuscript.

- can the authors give a full characterization of the spin states of left dot (the spin filter)? What is the g factor? What is the SOC strength? The authors claim that at 100 mT the Zeeman splitting is larger than V_L , which is larger than the induced gap. In contrast, the Zeeman shift of the ABS in the center dot is clearly much less than the gap at 100 mT.

Also, the claim of full spin polarisation depends on the nature of the states in the left dot, which is subject to the same SOC as the center section with the ABS.

The reviewer asks about the large difference in Zeeman energy for the QD and ABS at 100 mT, implying a significant difference of g -factor. This is important, considering the QD is only a spin filter when its Zeeman splitting is larger than V_L . We have added an extended data figure (ED9) to estimate the g -factors of QD and ABS:

Fig. ED9. g factors and QD level spacing. a. A measurement of I_L for the QD on the left side of the hybrid for varying B and dot gate V_{LM} . Horizontal lines estimate spacing between adjacent QD levels. A lever arm of 0.4 found from Fig. ED3 is used to convert gate values into units of energy, and calculate addition energies. The sloped lines estimate the effective g factor of the QD levels. b., c. A zoom in of Fig. 2b, c with dashed lines estimating the effective g factor of the QD

and lowest-lying ABS. d. The lines corresponding to the ABS effective g factor overlaid on top of the tunnel spectroscopy of Fig. 2a.

The $N \leftrightarrow N+1$ transition used for the down-filter (a) has an effective g -factor of 40. The $N+1 \leftrightarrow N+2$ transition used for the up-filter has an effective g -factor of 54. We also added effective g -factor lines to the QD (b, c) and tunnel spectroscopy (d). The opposite spin species for b and c is indicated by the green dashed lines and moves out of the bias window with a high g -factor. Taking the Zeeman splitting $\Delta E_Z = 2|E_Z| = g\mu_B B$, we find for panel b: $\Delta E_Z = 230 \mu\text{eV}$ for the ABS and $\Delta E_Z = 470 \mu\text{eV}$ for the QD, at 100 mT. For panel c, we find $\Delta E_Z = 238 \mu\text{eV}$ for the ABS and $\Delta E_Z = 575 \mu\text{eV}$ for QD at 100 mT. This confirms that we can polarize the QD by moving the opposite spin species out of the bias window well before driving the singlet-doublet transition.

In addition, the reviewer is concerned about the spin polarization of the spin-filter dot, as SOC could in principle lead to spin mixing thus diminishing the usage of the QD as a spin filter. To quantify the above, we note that for the QD, SOC of InSb acts as a matrix element between different orbital levels of opposite spin.

This means that a pure spin state gets mixed with another one by a proportion of $\frac{\langle H_{SO} \rangle}{\delta - \Delta E_Z}$, where δ is the level spacing and $\langle H_{SO} \rangle$ the magnitude of the matrix element (Hanson, et al., *Reviews of Modern Physics* 79.4 (2007): 1217). This corresponds to an avoided crossing of magnitude $2\langle H_{SO} \rangle$ between QD levels of opposite spin of different orbitals in energy, when $\delta = \Delta E_Z$. Comparing the two resonances that come closest in panel a, we find $E_Z = 3.1 \text{ meV}$ at 1 T and $\delta = 3.9 \text{ meV}$. Typically, $\langle H_{SO} \rangle$ for neighboring QD orbitals in these systems is seen to be around 200 to 400 μeV in our and other groups' previous works (see Ref 9 and the references therein), making the fraction of $\frac{\langle H_{SO} \rangle}{\delta - \Delta E_Z}$ small for the range of the field used in this experiment.

Finally, we note that the SOC strength in the QD and hybrid can in principle differ due to absence/presence of the superconducting contact. E.g., in our previous work (Ref 9), we found SOC in the hybrid structure can be stronger than InSb QD for certain ABS levels, possibly due to the large band-bending electric field at the semiconductor-superconductor interface.

minor points:

- Ref. 9 is missing bibliographic data

We have added the missing bibliographic data

- ED8 was probably measured with only one of the gates on the left dot operated to get a simple tunnel junction. This should be mentioned explicitly in the caption, otherwise it is confusing in the context of the other experiments.

Indeed, Fig ED8 was measured when both sides of the hybrid were tuned to be tunnel junctions. We have added a statement about the left side being a tunnel junction to Fig ED8 caption:

The gates forming a QD to the left of the hybrid are reconfigured as a tunnel junction to measure non-local conductance.

Reviewer #3 (Remarks to the Author):

In this paper, the authors implement spin-filtering experiments of the Al-InSb nanowires. The proximized region of the InSb nanowire is studied by tunnel spectroscopy. They reproduce the well-

known singlet-doublet transition invoked by the Zeeman splitting reported in similar nanowire experiments in the literature. An important progress is that they use the quantum dot adjacent to the proximitized region as a spin-filter probe under the magnetic field. Thanks to the QD spin probe, they succeed in the evaluation of the spin polarization of the excitation from the singlet or doublet ABSs. Consequently, they find the perfect spin polarization despite the spin-orbit interactions in the nanowire. In addition, they address relaxation from the excited state by switching the nonlocal probe. Although the spin-filter experiments have already been reported in the STM experiments as cited in the manuscript, their data quality seems much better than the previous STM experiments. In addition, the blockade of the relaxation of the ABS seems important to establish ABS physics. Their device structure is expected to be a platform for detecting and controlling the Majorana Fermions, which is currently attracting much attention, and the authors mention the relevance of the Majorana Fermions in their discussion of the magnetic field dependence.

The paper provides useful and valuable information for those who study the physics of superconductor-semiconductor junctions. However, I have several questions and comments on the results and discussion before I recommend the publication.

We thank the reviewer for the critical review and the positive assessment of the manuscript.

I have the following comments:

1. An electrical circuit in Fig. ED1 should appear in the main manuscript instead of Fig. 1d otherwise it seems difficult to understand what they measure.

We thank the reviewer for this comment. To address this issue and improve the readability of our manuscript, we have now added to the SEM image of our device and the attached electrical circuit.

2. They discuss the possible spin-flip process which can degrade the spin-filtering. The spin-flip can happen because the effective field direction of the SOI is perpendicular to the external field. They propose that the energy level spacing is large (compared to what?) enough to suppress the spin-flip, resulting in the perfect spin polarization. I cannot understand why this large spacing can prevent spin-flip tunneling. Even if the large energy splitting in the QD and the ABS, the spin-flip seems to be invoked by the SOI between the up-spin state in QD and the down-spin state of the ABS. A more detailed discussion is necessary.

The issue raised by the reviewer is indeed an important one that was under-addressed in the first version of the manuscript. In the main text of the manuscript, we report on complete spin blockade in the tunneling between a spin-polarized QD and a spin-polarized ABS. The observation of such a blockade indicates that the spins of the ABS and the QD are collinear, and, as pointed by the reviewer, that the spin of the electron tunneling between them is preserved.

The purity of the spin of the QD is guaranteed by having a large level spacing compared to the spin orbit interaction strength (see reply to the first comment by reviewer 2 for detailed discussion). The polarization of the ABS is one of the results of this work.

To better address the question of the spin-flip tunneling, we find it important to inform the readers and reviewer of other data measured with the same device when the tunnel barrier separating the QD and the ABS (V_{LI}) was set to different values.

The experiments shown in the main text were conducted with $V_{LI} = 135$ mV, a value which nearly pinches off the transport between the QD and the ABS. However, prior to these measurements, we also measured using a more transparent barrier setting of $V_{LI} = 168$ mV, which shows incomplete spin blockade. This dataset is shown below. (The higher tunneling rate incidentally also lead to significant inelastic tunneling visible in the data.) Higher tunneling rate is known to aid the lifting of spin blockade via increasing spin-orbit hybridization

between the singlet (0,2) and triplet (1,1) states (Nadj-Perge et al, PRB 81, 201305). Furthermore, since we expect that such small variation of the barrier voltage will not affect the ABS or the QD levels significantly, we attribute the appearance of a leakage current at high tunneling rate to spin-flip tunneling, as the referee suggests. Thus, a good measure of the spin polarization of the ABS can only be obtained when spin-flip tunneling is suppressed by tuning the tunnel barrier to be as opaque as possible. We present this measurement in ED10 and add the following discussion to the main text:

The presence of spin-flip tunneling, induced by the spin-orbit interaction, could result in the lifting of the observed blockade [45]. Previously, such spin-flip tunneling in InAs-based double quantum dots was modulated by controlling the barrier separating the QDs [46]. In Fig. ED10 we present spin-polarized spectroscopy taken with a more positive tunnel gate value V_{LI} , showing a small lifting of the spin blockade. Since we do not expect the change in the tunnel gate voltage to significantly affect the QD or the ABS levels, we interpret the partial lifting of the spin blockade as resulting from spin-flip tunneling.

Fig. ED10. Spin-polarized quantum dot spectroscopy of an ABS for different gate settings. All main text figures were obtained for $V_{LI} = 135$ mV, while here we set $V_{LI} = 168$ mV. a., b. I_L vs μ_{QD} and V_H using the QD as a \downarrow -filter (panel a) and \uparrow -filter (panel b) for $B = 400$ mT. Peaks in I_L can be seen for both negative and positive μ_{QD} , indicating incomplete spin-polarization. We further see signatures of inelastic tunneling compatible with the higher tunneling rate between the QD and the ABS [64]. c. I_L vs μ_{QD} and B at using the QD as a \downarrow -filter for $V_H = 356$ mV. We observe incomplete spin-polarization as in panels a and b.

We further note that the data shown in ED7, taken with device B, also shows incomplete spin polarization. There we attribute the lack of spin-blockade to imperfect polarization or non-collinear spins of both the QD and the ABS and cannot rule-out further contribution from spin flip tunneling. See further discussion of device B in the reply to the comment below.

3. Related to the above question, I want to know the differences between this device and their previous device [9]. Why does the spin-flip tunneling happen in the previous device?

As the reviewer points out, the device reported on in Ref [9] and the device reported on in this work share very similar geometry, but seem to report on contradictory results, namely in Ref [9] we show how the spin-precession induced by spin-orbit coupling allows us to couple states with seemingly incompatible spin polarization whereas in this work we report on a complete spin blockade.

The difference lies in the physical properties of the QD/ABS levels investigated and tunnel barrier transparencies, which are selected or set with different goals in mind. In this work, our aim was to use the QD as a non-invasive spectroscopic tool. To do so, we tuned the voltages applied to the finger gates, and especially to the barrier separating the QD and the ABS, in a way that gives the sharpest linewidth and minimized inelastic tunneling. This is done by quenching the transport between the QD and the ABS. In the case of Ref. [9], the purpose of the experiment was to measure crossed Andreev reflection and elastic co-tunneling between two QDs residing on both sides of the ABS. This experiment requires larger coupling between both QDs and the ABS. We tuned the barriers between the QDs and the ABS to be more transparent than as reported in this experiment (the current measured when the QD was resonant with the quasi-particle states above the gap was on the order of 1nA, compared with 100 pA in this work). As we discussed above, the larger tunneling rate there enhances spin flipping.

A second difference is mentioned above in reply to referee 2, namely the amount of spin precession in the QD-ABS-QD geometry is likely notably higher than QD-ABS owing to the larger total distance separating the spins and other microscopic differences between the ABSs investigated.

A third factor may also contribute to the strong spin blockade effect here. In Ref 9, the admixture of opposite spins into the QD eigenstates due to SOC played a small but nonetheless existent role in the lifting of spin blockade. In the current device, even this contribution is further suppressed by the larger level spacing (2 to 3 times than Ref 9). We again refer to the reply to referee 2 for details.

In addition, we performed spin-polarized spectroscopy on the device of reference 9 and present them in ED7. This required tuning the device away from the conditions reported in Ref. [9] (mostly increasing the tunnel barrier between the measuring QD and the ABS to quench the tunnel rate). These measurements show incomplete spin polarization, which partly arises from the lack of complete spin polarization in the QD itself and possibly from the presence of spin-flip tunneling between the QD and the ABS.

We have significantly modified the text accompanying ED. 7 to summarize this discussion and added two panels showing the level spacing and the field evolution of the QD. The caption of ED7 now reads:

Fig. ED7. Spin-polarized spectroscopy of a second device. Measurements conducted on the device reported on in Ref. [9], which has an Al length of 180 nm, with an additional 2 °A of Pt grown at 30°. Here, the QD formed on the right side of the ABS served as the spin-filter for the spin- polarized spectroscopy. Compared to Ref. [9], here the barrier between the spin-polarized QD and the ABS was tuned to be much higher, reducing the transport between the QD and the ABS, and reducing the amount of spin-flip tunneling. **a.** The current through the QD I_R as a function of V_{RD} , for fixed $V_R = 1$ mV. The two resonances observed around $V_{RD} = 790$ mV and $V_{RD} = 798$ mV serve as our \downarrow and \uparrow spin filters respectively. Note the additional resonance at $V_{RD} = 808$ mV, that we attribute to the next orbital level. Spin-orbit coupling on the QD can give rise to spin mixing decreasing the efficiency of the \uparrow spin filter. **b.** The current through the QD I_R as a function of V_{RD} and B, for fixed $V_R = 0.3$ mV. The two resonances evolve with the expected trajectory showing their spin polarization. We determine E_c and g_{eff} using a lever arm of 0.34 for the \downarrow -filter and 0.37 for the \uparrow -filter. These are also used for panels c-f. **c., d.** I_R vs μ_{QD} and B using the QD as a \downarrow -filter (panel a) and \uparrow -filter (panel b). The QD acts as a spin filter for $B > 75$ mT, indicated by the blue line. Below the blue line, both QD spins are within the bias window and participate in transport. We note that the \downarrow filter shows the expected behavior as discussed in the main text, with some leakage current resulting from spin-flip tunneling or spin-orbit effect in the ABS itself. The \uparrow filter shows little signs of polarization. We attribute this to strong spin mixing within the QD making it a poor spin-filter. **e., f.** I_L vs μ_{QD} and B using the QD as a \downarrow -filter (panel c) and \uparrow -filter (panel d).

4. I recommend representing their gate conditions (at least V_H and V_{RO}) in the captions for all the figures (or directly in the figures).

We thank the reviewer for the suggestion and have added the relevant gate voltages to the text.

5. I could not find the discussion of Figs. 2d and e in the section of "Zeeman-driven singlet-doublet

transitions". Are they necessary in the main manuscript? For the discussion of relaxation, Figure 4 is sufficient for me.

We thank the reviewer for pointing out our lack of discussion of Fig 2 d and e. We have added a small paragraph stating that the nonlocal current indicates that the ABS is extended and couples to both normal leads. We feel that this is important for understanding the relaxation process later on in Fig 4. Furthermore, showing the non-local current sign reversal in Fig 2 allows us to relate our experiments to the existing work [8,10-13,45].

Thus far, we have focused on the excitation of the ABS. To complete a transport cycle, the ABS must also be able to relax back to the ground state. This can be done by either emitting or accepting electrons from the right lead through the tunnel junction. As a consequence, finite QD-current I_L is generally accompanied by finite I_R at the corresponding energy and field, as shown in fig 2d, e. We observe the same sub-gap features as in panels b,c confirming that these are extended ABSs that couple to both normal leads. Upon crossing the singlet-doublet transition, the ABS relaxation requires an opposite direction of electron flow, giving rise to the sign switching of I_R at $B = 0.3$ T [8, 10-13, 45]. The precise sign of I_R depends on μ_H as discussed in more detail below.

6. They discuss the zero energy state as a quasi-Majorana state. However, I am very confused about the definition of the "quasi-" Majorana state. Do they want to discuss the interference of the two Majorana at both edges? If so, they need to represent the interference pattern by changing the magnetic field or similar results. Otherwise, they should not use the "Majorana" state as their interpretation.

The reviewer points out that they expect two Majorana edge modes to interfere once their wavefunctions overlap, giving rise to the phenomenon known as "Majorana oscillations" with increasing magnetic field. Since we do not show such a pattern, the reviewer is concerned that the use of the term quasi-Majorana is improper. To explain our intentions, we'd first like to clarify the term "quasi-Majoranas" as it was defined in Vuik et al (Vuik et al., *SciPost Phys* 7, 061 (2019)). In their theory paper, an ABS forms at zero energy that is localized at a smooth barrier potential, as in other works (see e.g., Kells et al. *PRB* 86, 100503, 2012, Avila et al., *Commun Phys* 2, 133 (2019), or Pan et al., *PRB* 104, 054510 (2021)). In this scenario there is no interference between Majorana zero-modes at opposite edges. The two spin parts of this localized ABS separate spatially such that one spin-part couples strongly through the tunnel barrier while the coupling for the other spin part is strongly suppressed. This is why one could probe only one spin-part of this localized ABS. Vuik et al. noted that one spin-part of an ABS at zero energy is equivalent to a Majorana zero-energy state although not protected by a topological phase, hence they coined the name "quasi-Majorana". Quite recently, the spin projection of these states was shown to be polarized along the direction of the applied magnetic field (Tian et al. *Results in Physics* 26 (2021): 104273).

However, following the comment made by the reviewer, we recognize the possible confusion that the term quasi-Majorana might create. To avoid unnecessary confusion, we will use a term less prone to misinterpretation - "persistent zero-energy conductance peaks". This term also covers zero-energy states that can occur in proximitized quantum dots due to the interplay between the spin-orbit coupling, the size of the quantum dot and the precise tuning of the chemical potential (Reeg et al. *PRB* 98, 245407 (2018)). In ED 8 we show spectroscopy of our device at a different value of V_H , resulting in a zero-energy conductance peak that does not stick to zero. The sticking of the states to zero energy is related to spin texture of the wave-function (Junger et al., *PRL* 125, 017701 (2020)), whose direction of spin polarization is spatially varying. Our measurements, showing a persistent zero-energy state with complete spin-polarization at one end of the QD are consistent with this interpretation. Because an ABS with said spin texture and quasi-Majoranas result in comparable signatures, we will refer to our observation as a persistent zero-bias conductance peak.

To address this, we replace the text discussing the persistent zero-bias conductance peak with the following:

At higher fields ($B > 0.6$ T), we observe another low-energy ABS in tunneling spectroscopy. Above $B \approx 0.75$ T, this ABS sticks to zero energy and is completely down-polarized (Fig. 2h). While a persistent, spin-polarized zero-bias peak is a predicted signature of Majorana bound states [47–49], the short length of our hybrid section excludes this interpretation [50]. The interplay of spin-orbit coupling and confinement is a known mechanism for the formation of persistent zero-bias conductance peaks in QDs coupled to superconductors, given precise tuning of the QD chemical potential [26]. Such states develop a spin texture that has a global vanishing magnetic moment but is locally spin-polarized [51]. Our observation of spin-polarized persistent zero-bias peak is consistent with this interpretation. We emphasize that this state is fine-tuned using V_H (see Fig. ED8 where for different value of V_H the zero-bias peak does not persist over a large range of B) and further study is required to fully characterize such states

7. They should cite the original theory papers of the Yu Shiba Rusinov states.

We thank the reviewer for this comment and have added these citations when introducing these states in our manuscript.

8. I expect that the sum of the spin-polarization of the excited ABS (for example in Fig. 3g) should be zero. They have better exhibit the analysis results and discuss the reason if the sum at a certain region is not zero.

The reviewer is surprised that the higher energy ABSs close to the gap edge are spin-polarized in, for example, Fig. 3g. We believe this is a byproduct of using quantum dots for spin-polarized tunneling. Close to the gap edge, electrons can tunnel from the QD into both the semiconductor and superconductor density of states, due to the finite QD broadening. As a result, we can no longer claim that the spin-polarization we measure corresponds only to the ABS. This makes the spin-polarization data more unreliable toward high energies. We have significantly changed the discussion around Fig. 3g accordingly:

Fig. 3g shows the corresponding spin polarization that was computed likewise to Fig. 2h. Transport at positive μ_{QD} is seen to be fully down-polarized in the singlet ground state. Likewise, states are up-polarized for negative μ_{QD} . Polarization appears as incomplete in the $|\downarrow\rangle$ ground state. This is a measurement artifact due to using QDs as spin filters. The up-, and down-filter transport peaks have finite broadening due to temperature and coupling to the leads. For ABSs close to zero, this can result in peaks at negative and positive μ_{QD} overlapping, as seen in the dark blue linecut of Fig. 3f, giving imperfect polarization.

A similar problem occurs for states close to the gap edge. We see that the higher energy ABS always appears down-polarized at positive μ_{QD} and up-polarized at negative μ_{QD} . Close to the gap edge, broadening results in tunneling into the Al density of states, in addition to the ABS. The inability to distinguish between tunneling into the metallic and semiconductor density of states makes the interpretation of spin polarization at higher energies unreliable.

9. They nicely address the relaxation of the excited ABS by controlling the nonlocal current I_R . This is impressive to me. In their results, there seems to be no leakage current, and the relaxation is completely blocked when V_{RO} pinches off the right nanowire region. I want to know more details about why the excited ABS are not leaked to the ground state. I think that the quasiparticle poisoning time decides the blockade time. Does this result indicate that the poisoning time is sufficiently long?

We thank the reviewer for their praise and share their interest in the relaxation mechanism. Our measured current noise floor is ~ 1 pA, which means we can only observe phenomena that relax the ABS faster than 100

ns. Parity lifetimes due to quasi-particle poisoning are estimated to exceed 10 μ s, the precise value depending on device specifics (Ménard, et al. *PRB* 100.16 (2019): 165307.). Considering this is slower than we can resolve, we do not expect to observe leakage current due to quasi-particle poisoning. We have added two sentences to the main text to point this out:

Our current noise level of ~ 1 pA gives a lower bound for the spin-relaxation time of ~ 100 ns. ABS parity lifetimes of 10 μ s or slower have been observed [53], which is significantly slower than we are able to resolve. We interpret this as meaning that there are no mechanisms in our system that can relax the ABS in less than 100 ns.

10. As they also mention, the spin-filtering experiments of the YSR or ABS have already been reported in the STM measurements. However, the previous results do not indicate the perfect spin-polarization if I understand correctly. Then if possible, please discuss why they can obtain the perfect polarization and what is the most critical ingredient or advantage in their device or setup.

We first note that the tunneling current from a ferromagnetic STM tip is only partially spin polarized, in contrast to the near 100% polarization in our QDs. For example, a Fe tip results in 40-45% spin-polarization, and a Cr tip about half of that (Wiesendanger, Roland. *Reviews of Modern Physics* 81.4 (2009): 1495.). We have added more STM references to the main text, including the work of Schneider et al where a YSR state STM tip was used for spin-polarized measurements (Schneider, Lucas, et al. "*Science Advances* 7.4 (2021): eabd7302.) We have added a small paragraph to our results section:

We note that our spin probe consists of only a single quantum state which becomes fully spin-polarized under the presence of a large Zeeman field. This is different from conventional spin-polarized tunneling in STM, where an exemplary Fe tip achieves 40-45% polarization [40]. In a more recent study using a YSR-state on a STM tip as a spin probe, the filtering mechanism using a single quantum state is comparable to that reported in this paper [33].

11. I think that the manuscript title is too simple. This title may mislead that this study is the first experimental work about the spin-filtering measurement of the ABS. I recommend changing their title (for example they need to clarify they use the superconductor-semiconductor hybrid devices).

We agree that the title should better reflect the role of the manuscript within the existing body of literature. We have therefore changed the title to "Spin-filtered measurements of Andreev bound states in semiconductor-superconductor nanowire devices".

Reviewers' Comments:

Reviewer #2:

Remarks to the Author:

My concerns have been addressed, and I recommend publication in Nature Communications.

Reviewer #3:

Remarks to the Author:

The authors thoroughly discussed and answered my concerns. I appreciate their discussion and revisions. Now I recommend this revised manuscript for Nature Communications.

Reviewer #2 (Remarks to the Author):

My concerns have been addressed, and I recommend publication in Nature Communications.

We thank the reviewer for their positive assessment and are excited to publish in Nature Communications.

Reviewer #3 (Remarks to the Author):

The authors thoroughly discussed and answered my concerns. I appreciate their discussion and revisions. Now I recommend this revised manuscript for Nature Communications.

We thank the reviewer for recommending our work for publishing.

We thank both reviewers for their meaningful and relevant questions. We appreciate the discussion and feel that it has improved the manuscript significantly.